# Cardiomyopathy Associated with Right Ventricular Apical Pacing-Systematic Review and Meta-Analysis

**DOI:** 10.3390/jcm11236889

**Published:** 2022-11-22

**Authors:** Andrzej Osiecki, Wacław Kochman, Klaus K. Witte, Małgorzata Mańczak, Robert Olszewski, Dariusz Michałkiewicz

**Affiliations:** 1Department of Cardiovascular Diseases, Bielanski Hospital, Centre of Postgraduate Medical Education, Ceglowska 80 Street, 01-809 Warsaw, Poland; 2Leeds Institute of Cardiovascular and Metabolic Medicine, University of Leeds, Woodhouse Lane, Leeds LS2 9JT, UK; 3Department of Gerontology, Public Health and Didactics, National Institute of Geriatrics, Rheumatology and Rehabilitation in Warsaw, 1 Spartanska Street, 02-637 Warsaw, Poland; 4Department of Ultrasound, Institute of Fundamental Technological Research, Polish Academy of Sciences in Warsaw, 5B Pawinskiego Street, 02-106 Warsaw, Poland

**Keywords:** artificial pacing, right ventricular pacing, left ventricular systolic function, heart failure

## Abstract

AIMS: Bradyarrhythmias are potentially life-threatening medical conditions. The most widespread treatment for slow rhythms is artificial ventricular pacing. From the inception of the idea of artificial pacing, ventricular leads were located in the apex of the right ventricle. Right ventricular apical pacing (RVAP) was thought to have a deteriorating effect on left ventricular systolic function. The aim of this study was to systematically assess results of randomized controlled trials to determine the effects of right ventricular apical pacing on left ventricular ejection fraction (LVEF). Methods: we systematically searched the Cochrane Central Register of Controlled Trials, PubMed, and EMBASE databases for studies evaluating the influence of RVAP on LVEF. Pooled mean difference (MD) with a 95% confidence interval (CI) was estimated using a random effect model. Results: 14 randomized controlled trials (RCTs) comprising 885 patients were included. In our meta-analysis, RVAP was associated with statistically significant left ventricular systolic function impairment as measured by LVEF. The mean difference between LVEF at baseline and after intervention amounted to 3.35% (95% CI: 1.80–4.91). Conclusion: our meta-analysis confirms that right ventricular apical pacing is associated with progressive deterioration of left ventricular systolic function.

## 1. Introduction

Pacemaker implantation was a ground-breaking innovation in the history of medicine. Artificial pacing is the only successful treatment for patients with life-threatening bradyarrhythmias such as atrio-ventricular second/third degree blocks or symptomatic sinus node dysfunction. From the very beginning, the pacing electrode was implanted in the right ventricle. Over the course of time, right ventricular pacing (RVP) was suggested by several studies to be associated with an increased rate of newly appeared left ventricular dysfunction or with progression of pre-existing left ventricular dysfunction [1,2,3]. In the absence of the most credible type of evidence regarding the left ventricular systolic dysfunction associated with RVAP, we aimed to systematically evaluate the current literature and conduct a meta-analysis of randomized controlled trials (RCTs) and prospective studies comparing the mid- and long-term effects of right apical ventricular pacing on left ventricular systolic function as measured by left ventricular ejection fraction (LVEF).

## 2. Methodes

### 2.1. Search Strategy

We systematically performed an electronic literature search of Cochrane Central Register of Controlled Trials, PubMed, and EMBASE databases from the time of database inception to October 2020 for studies evaluating the influence of RVAP on LVEF. We used search terms such as cardiomyopathy; heart failure; cardiac insufficiency; RV pacing; right ventricular pacing; artificial cardiac pacing; heart pacing; and artificial heart pacemaker during the search process. Reviews and reference lists of retrieved articles were hand searched for potentially relevant publications not previously identified in the database search. Our literature search was limited to prospective randomized trials published in peer-reviewed journals. All items resulting from these searches were reviewed at the title and abstract level and potentially eligible articles were reviewed in full text to assess eligibility. We also tried to obtain accessory information lacking in original articles through direct contact with the main authors.

### 2.2. Study Eligibility Criteria

We enrolled in our study prospective trials in which patients were at least 18 years old and underwent implantable cardioverter-defibrillator (ICD)/cardiac resynchronization therapy (CRT)/pacemaker implantation with pacing lead placement in the apex of the right ventricle. Because of statistical issues, cross-over trials needed to be excluded. Eligible studies had to report baseline LVEF and LVEF at the end of the follow-up or its change over the course of time. LVEF had to be assessed using echocardiography; trials in which LVEF was assessed using radionucleotide ventriculography were excluded from our analysis. Studies in which the RVAP percentage was absent or below 20% were excluded. Trials evaluating the effect of RVP after atrio-ventricular (AV) node ablation were also excluded. Studies in which a ventricular electrode was implanted in the right ventricular outflow tract or interventricular septum were not taken into consideration. Our analysis was limited to the groups of patients who were paced in synchronized atrio-univentricular mode or solely in univentricular mode (when the resynchronization device was implanted, the left ventricular lead had to be disactivated). We also limited our analysis to the trials in which the follow-up lasted at least 6 months.

### 2.3. Data Extraction

Independently, two investigators (A.O. and D.M.) extracted data from each eligible study, and potential disagreements were resolved by consensus. We documented the study characteristics (year of publication, study design, follow-up duration, and number of participants), patient demographics, and clinical characteristics (type of underlying cardiomyopathy, presence of hypertension, diabetes mellitus, coronary artery disease, or prior myocardial infarction with concomitant atrial fibrillation). We also extracted information about baseline LVEF, LVEF at the end of the follow-up, or differences between these two values.

### 2.4. Statistical Analysis

Pooled mean difference (MD) with a 95% confidence interval (CI) was estimated using a random effect model. Heterogeneity of the studies was determined using an inconsistency index I^2^ (0–100%) and between-study variance of true effects T^2^. An I^2^ value higher than 50% indicates substantial heterogeneity, and a value higher than 75% indicates high heterogeneity. A T^2^ > 0 is considered substantial. To assess the influence of each individual study on the overall result of the meta-analysis, a sensitivity analysis was performed, which consisted of removing the individual study from the calculations. Publication bias was evaluated using a funnel plot with the Begg–Mazumdar and Egger test and the Cochrane risk of bias tool for randomized trials (RoB 2). To assess the influence of individual studies on the meta-analysis, calculations were repeated excluding one of the studies. A *p*-value of less than 0.05 was considered significant. All statistical analyses were carried out using STATISTICA v.13.1 (Dell Inc. 2016, Tulsa, OK, USA).

## 3. Results

The systematic review and meta-analysis were performed according to the Preferred Reporting Items for Systematic Reviews and Meta-Analyses (PRISMA) statement for reporting systematic reviews and meta-analyses of RCTs. Our literature search identified 574 studies (Figure 1). After the exclusion of duplicates and non-relevant studies, 36 studies were retrieved for further full-text evaluation by reviewing study titles and abstracts. We further excluded three studies that were projected in cross-over manner [4,5,6], six trials that did not evaluate LVEF in the follow-up [7,8,9,10,11,12], seven studies in which information about RVAP was absent or RVAP was <20% [13,14,15,16,17,18,19], three studies in which patients were treated according to the ‘’ablate and pace” method [20,21,22], one trial with a short-term follow-up [23], one trial which was a prolonged follow-up of the so far included study, and one study because of statistical issues [24]. Finally, fourteen randomized controlled studies published in the period of 2008–2017 were included in this systematic review and meta-analysis.

Study and patient characteristics are summarized in Table 1. The eligible studies included a total of 885 patients. Study participants were predominantly male (ranging from 30% to 72.8% male), and their mean age ranged from 67.1 to 77 years. The apical location of the ventricular pacing electrode was defined in 13 studies; in the Block HF study, the pacing/defibrillating electrode was placed in the apex in the majority of patients (211/342). The mean follow-up in the included studies ranged between 6 and 89 months.

### 3.1. Risk of Bias

Using a revised Cochrane risk of bias tool for randomized trials (RoB 2), risk of bias of the included studies was estimated (Figure 2.)

### 3.2. Echocardiographic Changes

(a)LVEF. The mean difference between LVEF at baseline and after intervention amounted to 3.35% (95% CI: 1.80–4.91). A forrest plot of pooled differences is presented in Figure 3. There was substantial heterogeneity among the included studies: I^2^ = 72.1 (95% CI: 52.2–83.7) and T^2^ = 5.6 (95% CI: 2.4–11.0). The sensitivity analysis showed no significant changes in the overall result of the meta-analysis. When excluding individual studies from the calculations, the overall result was still above 3 and ranged from 3.01 (95% CI: 1.49–4.53) to 3.72 (95% CI: 2.19–5.26). No relationship has been found between effect sizes and standard errors: Begg–Mazumdar test; *p* = 0.625, Egger test; *p* = 0.775.

(b)LVESV. Six studies reported results regarding change in LVESV during the course of the trial. The overall mean difference in LVESV equalled −2.09 mL (95% CI: −5.30–1.13), indicating no significant change of LVESV after RVP. There was moderate heterogeneity across studies (I^2^ = 57.19%; 95% CI: 0.00–82.72%). The exclusion of individual articles did not significantly change the overall result. It was still statistically insignificant and ranged from −0.92 (95% CI: −4.12–2.28) to −2.91 (95% CI: −6.43–0.61).(c)LVEDV. Six studies reported results regarding change in LVEDV during the course of the trial. The mean difference in LVEDV was 0.45 mL (95% CI: −7.05–7.45), indicating a lack of RVP influence on LVEDV. There was substantial heterogeneity across studies (I^2^ = 80.89%; 95% CI: 58.88–91.12%). The exclusion of individual studies did not change the overall result significantly. It ranged from -0.89 (95% CI: −8.71–6.93) to 2.50 (95% CI: −2.62–7.62).(d)6 min walk test (6MWT). Six studies reported result of 6 min walk test both initially and at the end of the follow-up. The pooled mean difference was −26.93 m, which means that RVAP was associated with an increase in 6MWT after the intervention (Figure 4.). There was substantial heterogeneity across studies (I^2^ = 44.99%, 95% CI: 0.00–79.83%). The sensitivity analysis showed no significant changes in the overall 6MWT result. When excluding individual studies from the calculations, the range of values was from −21.04 (95% CI: −41.89–0.20) to −38.99 (95% CI: −61.14–16.85).

## 4. Discussion

To the best of our knowledge, this is the first study to analyze the influence of right ventricular apical pacing on LVEF in patients with dominant sinus rhythm and to comprise only prospective randomized trials. The result of our meta-analysis confirms that RVAP is associated with continuous deterioration of the left ventricular systolic function.

One of the most important factors predisposing individuals to LV systolic dysfunction is a substantial percentage of RVAP. Systolic function impairment measured by a shortening fraction in the setting of a high percentage of RVP (90%) was seen in the DDDR group in the study comparing AAIR and DDDR pacing in patients with sick sinus syndrome [39]. In the study conducted by Gierula et al., investigators reprogrammed pacemakers to minimize RVP in patients with avoidable RVP. After six months of follow-up, they obtained a mean of 49% reduction in RVP with simultaneous LVEF improvement at the level of 6% [40]. In the ANSWER trial, in which the SafeR strategy was implemented to reduce RVP in the general pacemaker population, it was shown that this tool was effective in RVP reduction in comparison to typical DDD pacing, although it did not have an overall effect on the composite primary end point (hospitalization for heart failure (HF), atrial fibrillation (AF), or cardioversion). However, it is worth mentioning that in this trial, HF hospitalizations showed a trend favouring SafeR [41]. Regarding the PreFER MVP trial, despite a reduction in RVP through the use of MVP (Managed Ventricular Pacing) mode, it did not lead to clinical benefits in terms of cardiovascular (CV) hospitalizations, death, or permanent or persistent AF [42]. Finally, Kiehl et al. concluded that, in populations with complete heart block and preimplant preserved LVEF, a right ventricular pacing burden of ≥20% was strongly associated with the development of PICM (Pacemaker induced cardiomyopathy) [43].

Another factor which might influence susceptibility to “pacing”-induced HF is the degree of left ventricular systolic dysfunction at the time of pacemaker implantation. In the DANPACE study, in which AAIR pacing was compared to DDDR pacing in patients with sick sinus syndrome (SSS) and preserved left ventricular systolic function (mean follow-up of 5.4 ± 2.6 years), no difference related to hospitalizations for HF, New York Heart Association functional class, or use of diuretics was found, despite a high percentage of ventricular stimulation in DDDR group (65 ± 33%) [11]. On the other hand, the study conducted by Albertsen et al. found LVEF deterioration in patients with preserved systolic function, a third-degree AV block, and DDDR pacemaker implantation, although this result did not yield change in functional status. Notably, in this trial, the ventricular electrode was implanted in the right ventricular outflow tract [44]. A similar result was reported by Crevelari; in her research, no hospitalization due to heart failure was observed. However, 23.5% of patients with RVP had over 10% LVEF reduction at the end of the follow-up [7]. According to the three meta-analyses, right ventricular non-apical electrode location ensures less harmful influence on LVEF in comparison with its apical location [45,46,47].

According to the DAVID (Dual-Chamber Pacing or Ventricular Backup Pacing in Patients With an Implantable Defibrillator) trial, the DDDR pacing mode appeared to be disadvantageous for patients with reduced EF (mean LVEF 40% or less), typical indications for an ICD, and no necessity for antibradycardia pacing [3]. The population of patients with reduced EF and prior MI was also investigated in the MADIT II trial, in which patients were either protected with ICD with antibradycardia ventricular backup pacing or continued conventional pharmacological therapy [48]. The authors of this study reported the concerning observation that patients treated with ICD had to be hospitalized more frequently due to new or worsened heart failure.

Additional insight into PICM (pacemaker-induced cardiomyopathy) came from retrospective analysis of the MOST trial. Sweeney et al. suggested that pacemaker-induced cardiomyopathy is a complex medical entity dependent on “play” between substrates and promoters. Substrates were described as specific clinical and physiological variables present at the pre-implantation stage, such as atrial rhythm, AV conduction, ventricular conduction, ventricular function, symptomatic heart failure, and prior myocardial infarction (MI). Promoters were directly associated with pacemaker implantation and postimplantation parameters, including ventricular desynchronization (paced QRS duration and cumulative %VP) and AV desynchronization (pacing mode). Surprisingly, not only prolonged postimplantation paced QRS duration but also prolonged postimplantation spontaneous QRS duration increased the risk of heart failure development, especially in patients with history of MI and low EF [49]. In our analysis, we were not able to systematically assess the influence of postimplantation paced QRS duration on LVEF because of insufficient data.

We suspect that pacing duration has substantial importance for PICM development. The longest duration of follow-up in the trials included in our meta-analysis was 89 months. In the long-term follow-up of patients included in the PACE trial, which lasted almost 5 years, investigators observed a significant LVEF reduction (62.0  ± 6.3% vs. 53.2  ± 8.2%) with a simultaneously maintained high RVP percentage (94.5  ±  19.5%) [24]. The follow-up duration in most of the aforementioned trials lasted up to 5 years [11]. Therefore, we cannot exclude that systolic dysfunction appears or accelerates after this period of time.

Our analysis showed no relationship between RVP and changes in LVESV or LVEDV.

The result of the 6MWT analysis was surprising is. Almost all studies included patients with preserved EF, except for one (Kaye et al.) in which patients’ LVEF was at least 40%, but no clinical signs and symptoms were observed. Four studies (Albertsen, Kaye, Yu, and Gong) reported deterioration of LVEF during the follow-up. Intuitively, a dependence between LVEF and the distance covered in the 6MWT would be expected. However, according to the literature, there is no correlation between systolic function and distance covered in the 6MWT in patients with heart failure [50,51]. A partial explanation for abovementioned result is chronotropic incompetence and atrio-ventricular conduction impairment, which were primary indicators for antibradycardia pacing in the analyzed studies. Sinus node dysfunction and atrio-ventricular conduction disabilities might be responsible for an insufficient cardiovascular response to exercise tolerance tests.

Ordinary right ventricular pacing is less expensive and less time consuming in comparison with innovative types of pacing such as resynchronization therapy or conducting system pacing. Whenever unfavourable effects of RVP are more probable (prospective high percentage of ventricular pacing, symptomatic heart failure, or young patient age), strategies other than RV apical pacing should be considered.

### Limitations

In the study conducted by Gierula et al., the patients included in the trial were already pacemaker dependent (unavoidable RVP > 80%), and they had been paced for at least 10 years before pacemaker replacement. Based upon this information, we cannot estimate the real influence of RV pacing on left ventricular systolic function, as their LVEF might have already been impaired due to substantial pacing percentages [28]. Our focus was to evaluate apical pacing, but we included one study in which the pacing/defibrillating electrode was located within the right ventricle but not in the RV apex in the minority of participants (131/342), which might influence the results of our study [25]. In studies conducted by Leong, Cano, Lewicka-Nowak, and Molina, baseline echocardiography examinations were made on active pacing, which might have influenced initial LVEF measurements [32,33,37,38]. Significant differences in follow-up duration among studies (from 6 months to 89 months) also emerged as a limitation of this study. We cannot completely exclude that LVEF at the end of the follow-up was biased by conditions having significant influence on left ventricular systolic function (such as myocardial infarction or myocarditis), but it would be rather unusual that these co-morbidities would have changed results of 14 randomized controlled trials and obscured real effects of RVAP on LVEF. We could not accurately analyze pre-implantation data influencing LVEF deterioration because of their incompleteness.

## 5. Conclusions

Our meta-analysis confirmed that right ventricular apical pacing is associated with progressive deterioration of left ventricular systolic function. The association between RVP and LV systolic function deterioration was particularly observed in patients with high RVP percentage. Before pacemaker implantation, baseline patient characteristics always need to be collected in order to properly qualify patients for intervention and avoid adverse effects of apical pacing in the future.

## Figures and Tables

**Figure 1 jcm-11-06889-f001:**
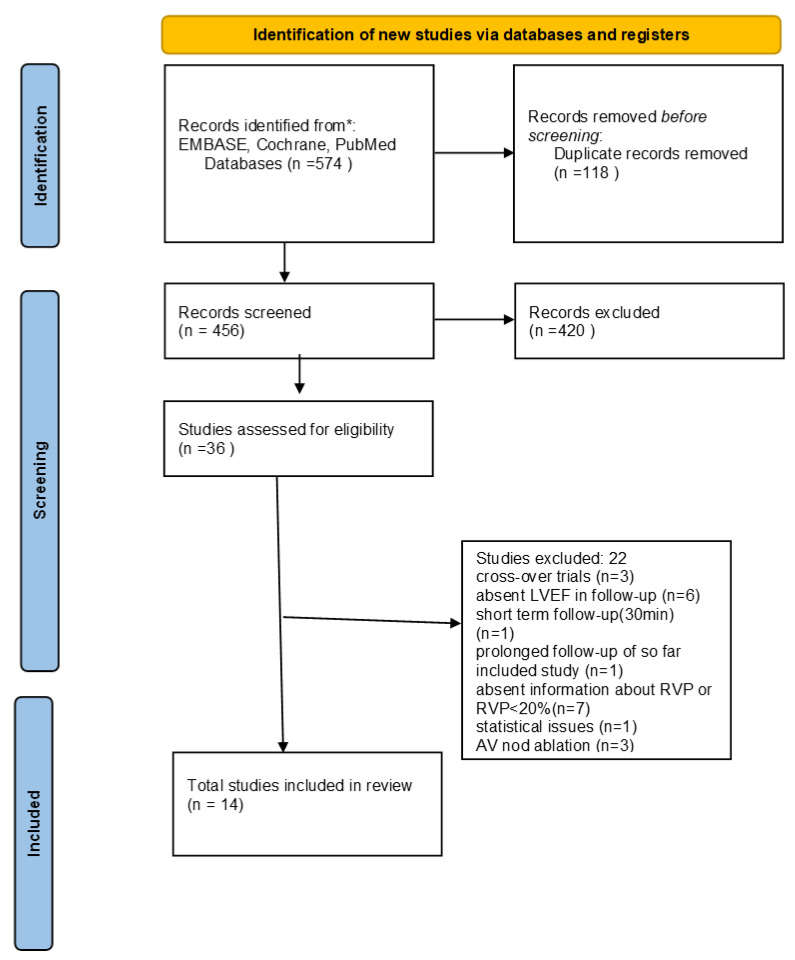
PRISMA flow diagram.

**Figure 2 jcm-11-06889-f002:**
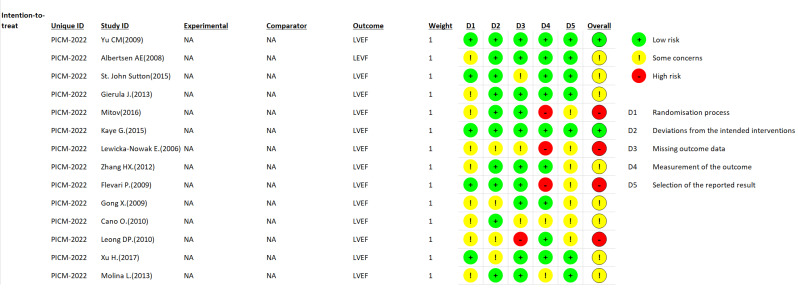
Risk of bias summary for the included studies. Green indicates low risk of bias. Red indicates high risk of bias. Yellow indicates some concerns [25,26,27,28,29,30,31,32,33,34,35,36,37,38].

**Figure 3 jcm-11-06889-f003:**
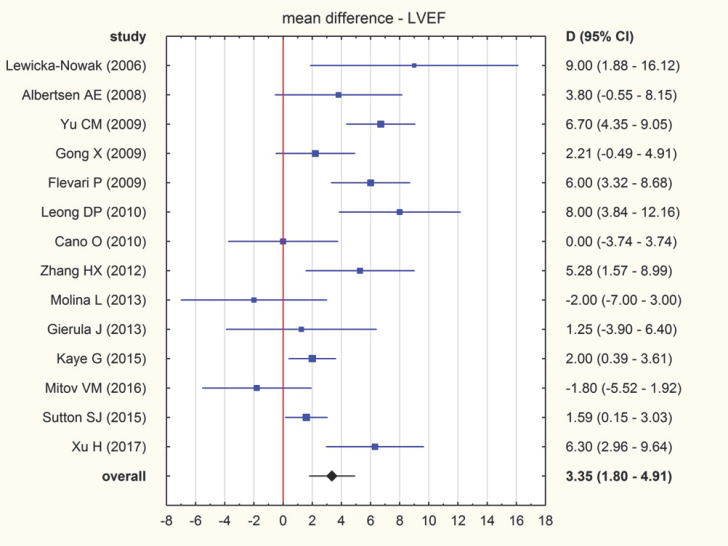
Forrest plot illustrating pooled mean difference (MD) with a 95% confidence interval (CI) of LVEF (%) [25,26,27,28,29,30,31,32,33,34,35,36,37,38].

**Figure 4 jcm-11-06889-f004:**
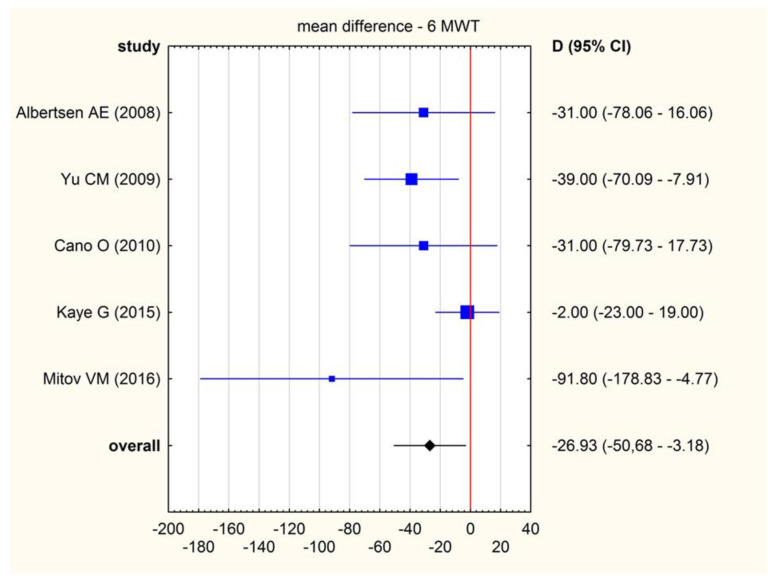
Forrest plot illustrating pooled mean difference (MD) with a 95% confidence interval (CI) of 6MWT (meters) [26,27,30,31,37].

**Table 1 jcm-11-06889-t001:** Studies and patient characteristics.

Author (Year)	Title	Reason for Cardiac Implantable Electronic Device (CIED) Implantation	Primary Pacing Mode in Analysed Group	Right Ventricular Lead Location in Analysed Group	Study Design	Observation Period (Months)	Subjects, n	Men, n (%)	Age, Mean (SD) (Years)	RV Pacing (%)	Baseline LVEF (%) (SD)	LVEF in Follow-Up (%) (SD)	Baseline Mean QRS Duration, (ms)	Paced QRS Duration (ms)	Baseline LVEDD (mm)	Follow-Up LVEDD (mm)	Baseline LVEDV (mL)	Follow-Up LVEDV (mL)	Baseline LVESV (mL)	Follow-Up LVESV (mL)
St. John Sutton et al. (2015) * [25]	Reverse Remodeling with Biventricular Pacing	Atrio-ventricular block in patients with heart failure in NYHA functional class I-III and LVEF ≤ 50%	DDD(R)/VVI(R)	Apical position (211/342) Other than apical position within RV (131/342)	Multicenter, prospective, randomized, double-blinded trial	24	342	249 (72.8)	73.0 ± 10.6	98.6-AVB 3d–167 patients 97.8-AVB 2d- 108 patients 97-AVB 1d–66 patients	33.6 ± 9.2	−1.6%, (95% CI), SD 10,499	123 ± 30.8	NA	CRT-P RVP 53 ± 8 CRT-D RVP 54 ± 7	CRT-P RV 53 ± 7.9 CRT-D RV: 57 ± 8.27	189.2 ± 64.0	188.0 ± 66.8	127.9 ± 53.1	129.8 ± 58.7
Albertsen AE. et al. (2008) [26]	DDD(R)-pacing, but not AAI(R)-pacing induces left ventricular desynchronization in patients with sick sinus syndrome: tissue-Doppler and 3D echocardiographic evaluation in a randomized controlled comparison	Patients with sick sinus syndrome with either syncope or dizzy spell or heart failure and electrocardiographic abnormalities: sinus arrest > 2 s or tachy-brady syndrome with sinus pauses > 2 s or sinus bradycardia < 40 bpm in awake hours	DDD(R)	Apical position	Single-center, prospective, randomized trial	12	26	8 (30)	73 ± 13	66	63.1 ± +8	59.3 ± 8	NA	NA	NA	NA	NA	NA	NA	NA
Yu CM. et al. (2009) [27]	Biventricular pacing in patients with bradycardia and normal ejection fraction (PACE).	Patients with sinus node dysfunction or bradycardia due to advanced atrioventricular block and normal ejection fraction((45%)	DDDR	Apical position	Multicenter, prospective, double-blind, randomized trial	12	88	49 (56)	68 ± 11	97	61.5 ± 6.6	54.8 ± 9.1	107 ± 30	NA	NA	NA	73.3 ± 19.8	76.7 ± 22.5	28.6 ± 10.7	35.7 ± 16.3
Gierula J. et al. (2013) [28]	Cardiac resynchronization therapy in pacemaker-dependent patients with left ventricular dysfunction.	Patients with implanted pacemaker, unavoidable RV pacing > 80%, reduced LVEF < 50% listed for routine pacemaker replacement due to battery depletion	DDD(R)/VVI(R)	Apical position	Single-center, prospective, randomized, unblinded trial	6	25	16 (64)	77 ± 4	98.34 ± 3.47	41 ± 4 (37–45) 95% CI)	39.75 ± 8.27	159 ± 10	NA	49.2 ± 3	NA	NA	NA	NA	NA
XU H. et al. (2017) [29]	Early Right Ventricular Apical Pacing-Induced Gene Expression Alterations Are Associated with Deterioration of Left Ventricular Systolic Function. Dis Markers	Patients with complete atrio-ventricular block and preserved LVEF ≥ 50%	DDD(R)	Apical position	Single-center, prospective randomized controlled trial	24	30	17 (56.7)	67.1 ± 7.5	100	63.0 ± 5.4	56.7 ± 7.6	102 ± 11	154 ± 12	NA	NA	103.2 ± 11.4	NA	37.8 ± 5.1	NA
Mitov VM. et al. (2016) [30]	The Effect of Right Ventricular Pacemaker Lead Position on Functional Status in Patients with Preserved Left Ventricular Ejection Fraction	Patients with preserved EF ≥ 54% and indication for antibradycardia pacing	DDD(R)/VVI(R)	Apical position	Single-center, prospective, randomized trial	12	61	43 (70.5)	72.72 ± 9.4	68.55 ± 39.34	59.16 ± 10.43	60.96 ± 10.56	91.15 ± 20.33	151.34	NA	NA	NA	NA	NA	NA
Kaye, G. et al. (2015) [31]	Effect of right ventricular pacing lead site on left ventricular function in patients with high-grade atrioventricular block: results of the Protect-Pace study	Patients with persistent 2:1 atrio-ventrcular block or higher and sinus rhythm or permanent AF and heart block with LVEF ≥ 40% and no clinical signs of heart failure	DDD(R)/VVI(R)	Apical position	Multicenter prospective randomized, trial	24	120	73 (60.8)	73.7 ± 11.1	98 ± 11	57 ± 9	55 ± 9	NA	NA	NA	NA	NA	NA	NA	NA
Molina L. et al. (2013) [32]	Medium-Term Effects of Septal and Apical Pacing in Pacemaker-Dependent Patients: A Double-Blind Prospective Randomized Study	Patients with complete heart block with no evidence of severe heart failure (NYHA IV).	DDD(R)	Apical position	Single-center, prospective randomized double-blind trial	24	34	14 (40.3)	72 ± 12	≥98	52 ± 10	54 ± 11	NA	158 ± 29.5	50 ± 8	46.9 ± 6.2	70.6 ± 34.0	61.9 ± 22.2	35.6 ± 27.1	31.8 ± 20.7
Lewicka-Nowak E. et al. (2006) [33]	Right ventricular apex versus right ventricular outflow tract pacing: prospective, randomised, long-term clinical and echocardiographic evaluation.	Patients with indications for permanent pacing, who required VDD, DDD or VVI/R pacemaker implantation.	VDD/DDD/VVI(R)	Apical position	Single-center prospective, randomised trial	89 ± 9	14	7 (50)	76 ± 9	94 ± 13	56 ± 11	47 ± 8	NA	QRS duration at the initial examination- 154 ± 16 QRS duration at the ned of follow up- 178 ± 19	49 ± 6	49 ± 8	NA	NA	NA	NA
Zhang HX. et el. (2012) [34]	Comparison of right ventricular apex and right ventricular outflow tract septum pacing in the elderly with normal left ventricular ejection fraction: long-term follow-up.	Patients between 65 to 85 years of age with conventional pacing indications for permanent pacing, no clinical manifestations of congestive heart failure (HF) and chronic renal insufficiency; without diagnosed AF prior to pacemaker implantation	DDD(R)	Apical position	Single-centre prospective, randomised trial	31.5 (13–58)	32	18 (56)	75 ± 10	82.91 ± 13.32	59.5 ± 6.21	54.22 ± 8.73	106.25 ± 18.36	143.56 ± 12.90	47.16 ± 3.63	49.22 ± 5.16	NA	NA	NA	NA
Flevari P. et al. (2009) [35]	Long-term nonoutflow septal versus apical right ventricular pacing: relation to left ventricular dyssynchrony.	Patients with persistent first-degree AV block, a relatively long PR interval (PR > 280 ms), a sinus rate > 60 bpm, and intermittent second- or third-degree AV block.	DDD(R)	Apical position	Single-center, prospective, randomised, trial	12	15	9 (60)	72 ± 1.5	97 ± 5	49 ± 4.3 On intrinsic rhythm	43 ± 3.1 On paced rthythm	153 ± 5.1	171 ± 4.5	NA	NA	85 ± 4.9	96 ± 5.2	39 ± 4.0	43 ± 3.0
Gong X. et al.(2009) [36]	Is right ventricular outflow tract pacing superior to right ventricular apex pacing in patients with normal cardiac function?	Patients with high or complete atrio-ventricular block, LVEF > 50% and no clinical signs of congestive heart failure necessitating permanent pacemaker implantation	DDD(R)	Apical position	Single-center, prospective, randomized trial	12	44	25 (57)	70 ± 11	97.3	67.92 ± 6.38	65.71 ± 6.56	97.23 ± 8.89	177.14 ± 22.52	NA	NA	84.32 ± 22.05	78.45 ± 17.91	27.23 ± 9.54	26.70 ± 9.54
Cano O.et al. (2010) [37]	Comparison of effectiveness of right ventricular septal pacing versus right ventricular apical pacing.	Patients with an indication for permanent cardiac pacing because of atrioventricular block or sick sinus syndrome, with no sings of heart failure and LVEF ≥ 50%;	DDD/VVI	Apical position	Single-center prospective randomized, single-blind,	12	28	14 (50)	72 ± 10	88.4 ± 17.1	62.9 ± 6.3 On paced rhythm	62.9 ± 7.9 On paced rhythm	NA	NA	NA	NA	88.6 ± 24.3	79.5 ± 29.8	33.2 ± 12.9	30.1 ± 14.5
Leong DP. et al. (2010) [38]	Long-term mechanical consequences of permanent right ventricular pacing: effect of pacing site.	Patients with conventional indications for pacemaker implantation(SSS; AVB) without indications for cardiac resynchronization therapy.	DDD	Apical position	Double-center prospective, randomized trial	30 ± 12	26	16 (61)	77 ± 8	49 ± 42	60 ± 6 On paced rhythm	52 ± 9 On paced rhythm	NA	156 ± 21	NA	NA	NA	88 ± 39	NA	45 ± 26

* Discrepancies in baseline LVEF between an article witten by Curtis and paper written by Sutton, both of them being based on the same Block HF Study, derive from different data collection methods. Curtis used data reported by the sites on the patient history form. Sutton, on other hand, used data measured by the echo core lab. LVEF—left ventricular ejection fraction; LVEDD—left ventricular end-diastolic diameter; LVESD—left ventricular end-systolic diameter; LVEDV—left ventricular end-diastolic volume; LVESV—left ventricular end-systolic volume; NA—not available.

## Data Availability

Direct contact with main author, A.O., e-mail: mcosiek@gmail.com.

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
