# Peer review of "Cardiomyopathy Associated with Right Ventricular Apical Pacing-Systematic Review and Meta-Analysis"

_jcm, 2022, doi:10.3390/jcm11236889_

Round 1

Reviewer 1 Report

Osiecki A et al., report a meta-analysis of pacing-induced cardiomyopathy in 14 studies with apical pacing. The findings of the study are interesting and show that indeed standard RV pacing is associated with the decrease of EF in patients in which RV pacing burden is above 20%. Studies promoting the risk of pacing-induced CMP are important in the era of developing techniques that could potentially prevent EF deterioration. However, there are some issues that need to be addressed.

MAJOR

1. The reason not to include studies (patients) with RV septal pacing is not totally clear as there were no studies that showed any benefits of RV septal pacing compared to RV apical pacing, therefore both RV lead locations should pose a similar detrimental effect on LV function (EF). Although the reason for selecting only RV apical pacing is mentioned, it should be emphasized more in the discussion and limitations. Were there any analyses performed also with patients that had RV septal pacing?

2. Mean pre-implantation EF was below 50% in 3 included studies (Sutton et al., Gierula et al., and Flevari et al.). These patients represent a very different group of patients with probably more diseased hearts with different etiologies compared to patients with preserved EF. Based on the most recent guidelines and some major studies BLOCK-HF and BIOPACE (although never released) patients with initially decreased EF are at substantially greater risk of pacing-induced CMP. Where there any separate analysis of EF reduction in these group of patients?

3. EF is a very robust marker although most frequently used in studies addressing pacing-induced CMP. Was the method of assessing EF in included studies similar? 

MINOR

Line 268 - conduction system pacing is probably a better term than para-Hissian left bundle pacing.      

Author Response

We would like to thank the editor for his helpful advice and for giving us the opportunity to submit a revised manuscript. We would also like to thank the reviewer for taking the time to review our manuscript we feel that in responding to his excellent reviews our manuscript has significantly improved, reaching the exacting standards of novelty and reach for Journal of Clinical Medicine and its broad readership.

  1. The reason not to include studies (patients) with RV septal pacing is not totally clear as there were no studies that showed any benefits of RV septal pacing compared to RV apical pacing, therefore both RV lead locations should pose a similar detrimental effect on LV function (EF). Although the reason for selecting only RV apical pacing is mentioned, it should be emphasized more in the discussion and limitations. Were there any analyses performed also with patients that had RV septal pacing?

Ad 1. Thank you for this question.  According to the cited metaanalysis of randomized controlled trials, conducted by Shimony et al. in which non-apical location of the electrode was compared to  its apical location, non-apical pacing had less deteriorating influence on left ventricular sytolic function measured by LVEF. Aforementioned publication stated that difference between those two pacing locations was greater when follow-up period exceeded beyond 1 year and when primarily LVEF was compromised. Althought clinical significance of this phenomenon is uncertain.

We did not perform any additional statystical analysis of patients with non-apical location of the electrode as only  in one study (Sutton et al.) minority of patients (131/342) had  this type of pacing.

  1. Mean pre-implantation EF was below 50% in 3 included studies (Sutton et al., Gierula et al., and Flevari et al.). These patients represent a very different group of patients with probably more diseased hearts with different etiologies compared to patients with preserved EF. Based on the most recent guidelines and some major studies BLOCK-HF and BIOPACE (although never released) patients with initially decreased EF are at substantially greater risk of pacing-induced CMP. Where there any separate analysis of EF reduction in these group of patients?

Ad 2. This is an excellent and important question. Although we did not perform any additional analysis of trials in which LVEF was compromised at the pre-implantation stage because this group would consist of only 3 studies, what from our point of would not be suffuciently representative.

  1. EF is a very robust marker although most frequently used in studies addressing pacing-induced CMP. Was the method of assessing EF in included studies similar?

Ad 3. Thank you. LVEF in all qualified studies was assessed using echocardiography. One of the exclusion criteria was radionucleotide evaluation of LVEF.

  1. Line 268 - conduction system pacing is probably a better term than para-Hissian left bundle pacing.

Ad 4. Thank you for this excellent point.  Indeed. Conduction system pacing is as far better term than para-Hissian left bundle pacing.

Reviewer 2 Report

The authors performed a systematic review and meta-analysis, investigating in the longitudinal effect of right ventricular apical pacing on left ventricular ejection fraction. They identified 14 randomized clinical trials including 885 patients and right ventricular apical pacing was associated with reduced left ventricular ejection fraction at follow up. This topic is currently very important as alternatives to standard right ventricular pacing (such as conduction system pacing) are on the rise.

As far as I understood the study, the current title and abstracts are very misleading. I assumed from the title that the authors performed a meta-analysis, comparing right ventricular pacing with other methods of pacing regarding change in LVEF after follow up. However, they rather identified studies with a right-ventricular apical pacing arm and documented the development of LVEF in that arm. Unfortunately, this fact reduces the applicability of this study: LVEF may have been reduced during follow up by natural history of the patient or complications (such as acute myocardial infarction). However, this study does not answer the question if right ventricular apical pacing would be more harmful than other methods of pacing. I therefore recommend to either completely change the analysis or to adapt the abstract and title accordingly. To be called meta-analysis, the authors should identify studies with two treatment arms and compare both arms regarding outcome. They should not identify the natural course of LVEF of just one arm of the study. (I am sorry if I did not understand the study correctly – if so, please adapt the description of the methodology.)

Furthermore, the authors found only a minority of studies dealing with apical RVP. For example, Albakri et al found more studies and more patients (>1000; DOI: 10.15761/IMM.1000411).

Further comments regarding the manuscript: There are minor English errors. Citations are adequate (however, some important studies missing, see above). In general, statistics seems adequate (except for the comments written above).

Minor comments: 

-        Line 44: From the very beginning, THE pacing electrode…

-        Why were pace and ablate strategies excluded? Why did the authors include citation 25 although it did not include apical RVP?

-        The study of Gierula should be excluded because patients with a 10-year old pacemaker are a different population compared to remaining patients with recent pacemaker implantation.

-        Lines 303-360 should be adapted according to the Journal’s guidelines.

-        Title: Please remove „Title:“

Author Response

We would like to thank the editor for his helpful advice and for giving us the opportunity to submit a revised manuscript. We would also like to thank the reviewer for taking the time to review our manuscript we feel that in responding to his excellent review our manuscript has significantly improved, reaching the exacting standards of novelty and reach for Journal of Clinical Medicine and its broad readership.

  1. As far as I understood the study, the current title and abstracts are very misleading. I assumed from the title that the authors performed a meta-analysis, comparing right ventricular pacing with other methods of pacing regarding change in LVEF after follow up. However, they rather identified studies with a right-ventricular apical pacing arm and documented the development of LVEF in that arm. Unfortunately, this fact reduces the applicability of this study: LVEF may have been reduced during follow up by natural history of the patient or complications (such as acute myocardial infarction). However, this study does not answer the question if right ventricular apical pacing would be more harmful than other methods of pacing. I therefore recommend to either completely change the analysis or to adapt the abstract and title accordingly. To be called meta-analysis, the authors should identify studies with two treatment arms and compare both arms regarding outcome. They should not identify the natural course of LVEF of just one arm of the study. (I am sorry if I did not understand the study correctly – if so, please adapt the description of the methodology.)

Ad 1. Indeed. The title and abstract need to be corrected- whenever right ventricular pacing is mentioned, the apical location of the pacing electrode needs to be added- we are going to follow this comment.

Our primary focus was to study  apical pacing, not comparing it  to other form of antibradycardia pacing. We cannot completely exclude that LVEF at the end of follow up was biased by conditions having significant influence on left ventricular systolic function (like myocardial infarction or myocarditis) but it would be rather unusual that, aforementioned co-morbidities would have changed results of 14 randomized controlled trials and blurred real influence of RVAP on LVEF.        We add this to the limitations paragraph.

 How can we be sure that natural course of LVEF is its’constant significant deterioration? Study conducted by Lupon et al.( https://doi.org/10.1161/CIRCHEARTFAILURE.118.005652) as a matter of fact focused on HFpEF population but proofed that only minority of patients diagnosed with HFpEF dropped to HFrEF during follow-up that lasted 11 years.  However authors of this publication noticed  that smooth deterioration was observed (but LVEF was still above 50%). 

According to the definition of meta-analysis enclosed in Encyclopedia Britannica  in which meta-analysis is described  a tool used in statistics, ”to synthesize the results of separate but related studies. In general, meta-analysis involves the systematic identification, evaluation, statistical synthesis, and interpretation of results from multiple studies.” Based upon our study completely fulfill a meta-analysis criteria.  (Britannica (Stroup, Donna F. and Thacker, Stephen B.. "meta-analysis". Encyclopedia Britannica, 4 Oct. 2016, https://www.britannica.com/topic/meta-analysis. Accessed 6 November 2022.) What is more, we enclose links to the chapters from Cochrane Handbook for Systematic Reviews of Interventions which from our point view support methodology of our study (https://training.cochrane.org/handbook/current/chapter-10#section-10-5-2; https://training.cochrane.org/handbook/current/chapter-06#section-6-5-1 ).  To the best of our knowledge there are many metaanalyzes published in good medical journals that compare the measurements before and after the intervention, for example: (https://pubmed.ncbi.nlm.nih.gov/34628706/ ).  We hope these explanations have cleared up your doubts and you accept that our work is called a meta-analysis. 

  1. Furthermore, the authors found only a minority of studies dealing with apical RVP. For example, Albakri et al found more studies and more patients (>1000; DOI: 10.15761/IMM.1000411). Further comments regarding the manuscript: There are minor English errors. Citations are adequate (however, some important studies missing, see above). In general, statistics seems adequate (except for the comments written above).

Ad 2. Thank you for this excellent point.  Albakri included in his study publications with follow-up that lasted at least 2 months, in our minimal follow-up duration was set at 6 months (we wanted to assess medium and long term outcomes).  3 of included studies did not evaluate apical position of the pacing electrode- this being primary focus of our study.  Besides that,  inclusion and exclusion criteria in our study were more strict in comparison with Albakaris’.  But this is another trial supporting advantageous influence of non-apical RVP over apical RVP on left ventricular systolic function- we have add this publication to the references.

  1. Line 44: From the very beginning, THE pacing electrode…

Ad 3. Thank you for making this suggestion. We have  already corrected this sentence.

  1. Why were pace and ablate strategies excluded? Why did the authors include citation 25 although it did not include apical RVP?

Ad. 4. Thank you for pointing this out. Patients treated in ablate and pace strategy represent somewhat different population. Their initial systolic function might be compromised in tachyarrhythmic mechanism. After ablation and pacemaker implantation we potentially change one form of  reversible cardiomyopathy into another. Based upon we decided to exclude this population and leave it for separate analysis.

 Citation nr 25 is related to St. John Sutton publication and whole BLOCK HF trial- in this study majority of patients had right ventricular pacing electrode within its’ apex implanted(211/342). Apart from pacing/defibrillating electrode within right ventricle all patients had also left ventricular electrode implanted.

  1.  The study of Gierula should be excluded because patients with a 10-year old pacemaker are a different population compared to remaining patients with recent pacemaker implantation.

Ad 4. Thank you for this excellent question. Regarding Gierulas’ paper- we eventually decided to qualify this trial despite representing a little bit different population due to very high pacing percentage and its unavoidable influence on left ventricular systolic function. Apart from that we mentioned about limitations associated with this study in limitations paragraph.

  1. Lines 303-360 should be adapted according to the Journal’s guidelines.

 Ad. 6. We of course adapt it to the Journal’s guidelines.

  1.  Title: Please remove „Title:“

Ad. 7. Title page has been already corrected.

Round 2

Reviewer 2 Report

The authors improved the manuscript significantly. The following concerns are still open:

- I am still not convinced by the authors that it is a good idea to just look at only one treatment arm. In the last years, conduction system pacing has been evolved and septal pacing has always been discussed more effective in restoring left ventricular function. The study would massively benefit by comparison of more treatment arms (such as: LBBB pacing, HIS pacing, CRT, septal pacing). The authors mentioned the study 10.1002/ejhf.2360 which also resembles a meta-analysis. The major difference between the mentioned study and this study is that there is already a lot of evidence and data about apical ventricular pacing available, the method is used for many decades. On the other side, interatrial shunt is rather an experimental method.

- There is a formatting problem, I encourage to use the standard word file for preparation.

- The authors describe the study of Gierula as "little bit different population". However, in my opinion this resembles a completely different population. In that population pacing already had occurred for more than ten years and consequently a lot of remodelling has been made over the course of the years.

Author Response

  1. I am still not convinced by the authors that it is a good idea to just look at only one treatment arm. In the last years, conduction system pacing has been evolved and septal pacing has always been discussed more effective in restoring left ventricular function. The study would massively benefit by comparison of more treatment arms (such as: LBBB pacing, HIS pacing, CRT, septal pacing).

Ad. 1. We agree entirely with the expert reviewer and are grateful for their suggestions. However, the alternate lead positions are relatively new, with limited penetration and follow-up time. Moreover our study was designed simply to describe the effects of RV apical pacing on left ventricular function and not to compare lead positions on LV function. We entirely agree that this would be an interesting question, but would require an alternative hypothesis, and sampling process.

  1. The authors mentioned the study 10.1002/ejhf.2360 which also resembles a meta-analysis. The major difference between the mentioned study and this study is that there is already a lot of evidence and data about apical ventricular pacing available, the method is used for many decades. On the other side, interatrial shunt is rather an experimental method.

Ad. 2. Thank you for this comment. We entirely agree with the reviewer, although would point out that our comments were around the scientific approach taken rather than the topic.

  1. There is a formatting problem, I encourage to use the standard word file for preparation.

Ad. 3. Thank you for your comment. We have reviewed the manuscript carefully for font, paragraph and line spacing.

  1. The authors describe the study of Gierula as "little bit different population". However, in my opinion this resembles a completely different population. In that population pacing already had occurred for more than ten years and consequently a lot of remodelling has been made over the course of the years.

Ad. 4. We are grateful for this comment and agree entirely which underlies our first limitation.